# The A Body Shape Index Might Be a Stronger Predictor of Chronic Kidney Disease Than BMI in a Senior Population

**DOI:** 10.3390/ijerph182412874

**Published:** 2021-12-07

**Authors:** Bokun Kim, Gwonmin Kim, Eonho Kim, Jonghwan Park, Tomonori Isobe, Takeji Sakae, Sechang Oh

**Affiliations:** 1Department of Kinesiology, Silla University, Busan 46958, Korea; fabulousbobo79@gmail.com; 2Medical Research Institute, Pusan National University Hospital, Busan 49241, Korea; rlarnjsals47@gmail.com (G.K.); jpark@pnuh.co.kr (J.P.); 3Department of Physical Education, Dongguk University, Seoul 04620, Korea; eonkim@dongguk.edu; 4Faculty of Medicine, University of Tsukuba, Ibaraki 305-8575, Japan; tiso@md.tsukuba.ac.jp (T.I.); takejisakae@gmail.com (T.S.)

**Keywords:** anthropometric index, estimated glomerular filtration rate, geriatrics, obesity

## Abstract

The A Body Shape Index (ABSI) was recently introduced to quantify abdominal adiposity relative to the body mass index (BMI) and height. This cross-sectional study was performed to explore whether the ABSI is linked to chronic kidney disease (CKD) in older adults and compare the predictive capacity of the ABSI versus BMI for CKD. In total, 7053 people aged ≥ 60 years were divided into normal, mild, and moderate-to-severe CKD groups based on their estimated glomerular filtration rate (eGFR). The correlation of the ABSI with the eGFR and the differences and trends in the ABSI and BMI among the groups were analyzed, and the cutoff points for moderate-to-severe CKD were calculated. The association between the ABSI and CKD was stronger than that between the BMI and CKD. The ABSI had a better capacity to discriminate the CKD stage than did the BMI. The capacity of the ABSI to predict moderate-to-severe CKD was higher than that of the BMI and was more substantial in women than men. The ABSI cutoff points for CKD were ≥0.0822 and 0.0795 in men and women, respectively. In conclusion, the ABSI serves as a better index than the BMI for screening and detecting high-risk individuals with CKD.

## 1. Introduction

Chronic kidney disease (CKD) is closely associated with aging and increasing hypertension, type 2 diabetes, myocardial infarction, and cardiovascular mortality [1,2]. The prevalence of CKD in Korea is reportedly 8.2%, 13.7%, and >30.0% in people aged 20 to 34, 35 to 59, and ≥60 years, respectively [3,4]. Similarly, the prevalence of CKD in the United States is 5.7%, 8.9%, and 33.2% in people aged 20 to 39, 40 to 59, and ≥60 years, respectively [5]. With the rapid aging of the Korean population, it is expected that the number of patients with CKD will likely continue to increase [6]. Considering that CKD causes remarkable increases in medical costs, morbidity, and mortality in older adults [1,2], early screening, detection, and care are highly recommended for high-risk individuals with CKD to slow the progression to severe CKD.

Obesity is also a critical public health issue, and its association with CKD is broadly recognized [7,8,9,10]. Therefore, previous studies conducted analyses of CKD risk factors by universally adopting body mass index (BMI) as the obesity index [11]. However, BMI does not precisely reflect overall adiposity and does not distinguish visceral fat, which induces the onset of several health conditions. According to recent studies, increased fat mass percentage is significantly associated with CKD [12,13,14]. Accordingly, the assessment of fat mass-related parameters likely requires a measurement that is superior to BMI for the management of CKD. However, because expensive equipment or extensive measurements are necessary to assess fat mass-related parameters, a simple, inexpensive, and widely available method must be developed to screen and detect high-risk patients with CKD.

In 2012, a new and simple anthropometric index, the A Body Shape Index (ABSI), was introduced, which quantifies abdominal adiposity relative to BMI and height for predicting chronic disease [15]. To date, the validity and reliability of ABSI have been adequately verified in various nations and ethnicities [6,16,17,18]. ABSI was reported to outperform BMI in predicting all-cause mortality [15]. Additionally, ABSI reflects visceral adiposity independently of BMI and is a potent marker of arterial stiffening [19,20]. Furthermore, compared with BMI, it is a better predictor of the risk of diabetes [21]. Despite the outstanding performance of ABSI for predicting mortality and chronic disease, the association between ABSI and CKD remains unexplored to date.

We therefore hypothesized that ABSI can be used as an alternative index to BMI for predicting the prevalence of risk of CKD [22]. We conducted this cross-sectional study to explore whether the ABSI is associated with CKD in a population of older adults and compare the predictive capacity of the ABSI versus BMI for CKD.

## 2. Materials and Methods

### 2.1. Study Design and Participants

A database evaluating the general health and nutrition state and lifestyle of South Koreans from the Korea National Health and Nutritional Examination Survey (KNHANES), 2014–2018, was used. Of the 12,490 participants ≥ 60 years old from the KNHANES, 2014–2018, 7053 participants (3257 men and 3796 women) were selected for analysis. A flow diagram of participant recruitment is shown in Figure 1.

The Institutional Review Board of the Korea Centers for Disease Control and Prevention (KCDC) approved KNHANES (approval numbers 2013-07CON-03-4C, 2013-07CON-03-5C, 2018-01-03-P-A). Additionally, written informed consent was obtained from all of the participants before the study began, which was conducted in accordance with the ethical principles of the Declaration of Helsinki.

### 2.2. Anthropometric Indices

Waist circumstance (WC) and height were assessed to the nearest 0.1 cm. Body weight was assessed to within 0.1 kg with a digital electronic scale (JENIX DS-102; Dong Sahn Jenix Co., Seoul, Korea). Using this information, BMI was defined as body weight (kg)/height (m^2^), and ABSI was defined as WC × body weight^−2/3^ × height^5/6^ [15]. To compare these two anthropometric indexes (BMI and ABSI), the indexes were converted to a Z-score using the following equation: (assessment value − mean)/standard deviation [21].

### 2.3. Estimated Glomerular Filtration Rate (eGFR) and Biochemical Assessments

Blood samples were collected in the morning after a fast of ≥8 h. Creatinine was analyzed using the Jaffe rate-blanked creatinine assay and compensated at a certified laboratory (Seegene Medical Foundation, Seoul, Korea). eGFR was defined in accordance with the new Japanese coefficient-modified Modification of Diet in Renal Disease study equation: eGFR (mL/min/1.73 m^2^) = 194 × (serum creatinine)^−1.094^ × (age)^−0.287^ (×0.739 for women) [12,23,24]. The participants were separated into three groups based on eGFR tertiles: a normal (*N*) group for ≥60.0 mL/min/1.73 m^2^, a mild CKD (*M*_CKD_) group for 45.0–59.9 mL/min/1.73 m^2^, and a moderate-to-severe CKD (*S*_CKD_) group for <45.0 mL/min/1.73 m^2^ [12,23,24].

### 2.4. Statistical Analysis

Data are presented as mean ± standard deviation, and odds ratio and 95% confidence interval (*CI)*. An independent t-test or the Mann–Whitney U test was adopted to compare parameters between men and women. One-way ANOVA was used to compare mean anthropometry among the three groups. The Bonferroni post-hoc test was used when ANOVA showed significant differences (*p* < 0.05). In case of abnormal data distribution, the Mann–Whitney U test was used to analyze differences between the groups (*p* < 0.05). The Jonckheere–Terpstra test was used to determine trends in values between the three groups (two-tailed, *p* < 0.05). The Jonckheere–Terpstra test generates standardized statistics that indicate the strength of trends in parameters that increase or decrease across groups. For correlation between ABSI or BMI and eGFR, Pearson correlation was performed. Logistic regression was employed to evaluate the associations between Z-scores of ABSI or BMI and *S*_CKD_. The fully adjusted model was adjusted for potential confounders including education level, household income, smoking, drinking, handgrip strength, moderate to vigorous physical activity, and nutritional factors that are known or suspected to influence the association with CKD. SPSS software, ver. 20.0 (IBM, Inc., Armonk, NY, USA) was used for statistical analyses. The optimal cutoff points of ABSI and BMI in overall, male, and female participants for predicting CKD were derived from receiver operating characteristic (ROC) curves, the area under the ROC curves (AUC), sensitivity, and specificity. This analysis was conducted using MedCalc for Windows ver. 9.1.0.1 (MedCalc Corp, Mariakerke Ostend, Belgium).

## 3. Results

Table 1 shows the characteristics of study participants. The mean age of overall, male, and female participants was 69.5 years (*SD*: 6.3, 6.2, and 6.4, respectively), and no sex difference was observed. In overall, male, and female participants, the mean eGFR was 61.0 (*SD*: 12.6), 60.9 (12.5), and 61.1 (12.6), respectively; mean BMI was 24.2 (3.1), 23.9 (2.9), and 24.5 (3.3), respectively; and mean ABSI was 0.0806 (0.0042), 0.0811 (0.0038), and 0.0802 (0.0045), respectively. Significant differences in BMI and ABSI were observed between the sexes (*p* < 0.001), although the difference in eGFR was insignificant. Additional details on overall, male, and female participants are shown in the Supplementary Material (Appendix A).

The overall and sex-specific differences and trends according to eGFR category are shown in Table 2. In overall participants, the trend revealed a significant decreasing tendency in eGFR from the *N* to *S*_CKD_ groups (*SS*: −54.58; *p* < 0.001), whereas there was an increasing tendency in creatinine, age, body weight, WC, BMI, and ABSI (*SS*: 57.86, 23.22, 5.89, 11.41, 7.46, and 10.31, respectively; *p* < 0.001). Z-scores of BMI and ABSI showed the same tendency. Post-hoc tests demonstrated significant differences among the three groups in all examined parameters, except for height. eGFR decreased in order of the *N*, *M*_CKD_, and *S*_CKD_ groups, whereas creatinine, age, WC, ABSI, and Z-score of ABSI increased in order of these groups. Values for body weight, BMI, and Z-score of BMI in the *M*_CKD_ and *S*_CKD_ groups did not differ, although the values were significantly higher compared with the *N* group. Appendix A shows the results of analyses of additional blood parameters and several covariates in overall participants.

In men, the trend test revealed a significant decreasing tendency in eGFR across the groups from *N* to *S*_CKD_ (*SS*: −54.58; *p* < 0.001), while the opposite tendency was observed for creatinine, age, body weight, WC, BMI, and ABSI (*SS*: 23.61, 15.47, 5.76, 8.56, 6.38, and 6.48, respectively; *p* < 0.001). Z-scores of BMI and ABSI showed the same tendency as with BMI and ABSI. Post-hoc tests demonstrated significant differences among the three groups for all examined parameters, except for height. eGFR decreased in order of the *N*, *M*_CKD_, and *S*_CKD_ groups, whereas creatinine, age, WC, ABSI, and Z-score of ABSI increased in order of these groups. The values for body weight, BMI, and Z-score of BMI in the *M*_CKD_ and *S*_CKD_ groups did not differ, although they were significantly higher than those in the *N* group. Appendix A shows the results of analyses of additional parameters in men.

In women, the trend demonstrated a significant decrease in eGFR and height across the groups from *N* to *S*_CKD_ (*SS*: −58.50 and −2.45, respectively; *p* < 0.05). The opposite tendency was observed for creatinine, age, body weight, WC, BMI, and ABSI (*SS*: 57.81, 17.31, 3.15, 7.71, 4.43, and 7.92, respectively; *p* < 0.01). Z-scores of BMI and ABSI showed the same tendencies as with BMI and ABSI. Post-hoc tests revealed significant differences among the three groups in terms of all parameters. eGFR decreased in order of the *N*, *M*_CKD_, and *S*_CKD_ groups, but creatinine, age, WC, and ABSI increased in order of these groups. Height values in the *N* and *M*_CKD_ groups did not differ, although they were significantly higher than those in the *S*_CKD_ group. The values for body weight and BMI in the *M*_CKD_ and *S*_CKD_ groups did not differ. However, they were significantly higher than those in the *N* group. Z-scores of BMI and ABSI showed the same differences as with BMI and ABSI. Appendix A shows the results of analyses of additional parameters in women.

Table 3 shows overall and sex-specific Pearson correlations between ABSI or BMI and eGFR. In overall participants, the Pearson correlations between ABSI or BMI and eGFR were −0.146 and −0.101, respectively; in male participants, the correlations were −0.130 and −0.132, respectively; and in female participants, the correlations were −0.158 and −0.079, respectively (all *p* < 0.001).

The comparison of overall and sex-specific odds ratios for the association between the Z-score of ABSI or BMI and *S*_CKD_ is shown in Figure 2. Overall, male, and female participants were separated into tertiles based on Z-scores of the ABSI. In the unadjusted model, compared with the lowest tertile, the highest and middle tertiles had odds ratios of 2.93 (95% *CI*: 2.36–3.64) and 1.59 (1.26–2.01), 2.85 (2.09–3.88) and 1.59 (1.14–2.21), and 3.02 (2.23–4.08) and 1.60 (1.16–2.21), respectively, for *S*_CKD_. In the fully adjusted model, compared with the lowest tertile, the highest and middle tertiles had odds ratios of 2.72 (95% *CI*: 2.18–3.40) and 1.55 (1.23–1.96), 2.73 (1.99–3.76) and 1.62 (1.16–2.27), and 2.51 (1.84–3.43) and 1.41 (1.02–1.96), respectively. Regarding the Z-score of BMI, in the unadjusted model, compared with the lowest tertile, the highest tertile had odds ratios of 1.64 (95% *CI*: 1.34–2.00), 1.75 (95% *CI*: 1.31–2.34), and 1.54 (95% *CI*: 1.17–2.04), respectively, for *S*_CKD_. In the fully adjusted model, compared with the lowest tertile, the highest tertile had odds ratios of 1.67 (95% *CI*: 1.36–2.05), 1.96 (95% *CI*: 1.46–2.64), and 1.37 (95% *CI*: 1.03–1.82), respectively, for *S*_CKD_.

Overall participants, n = 7053; male participants, n = 3257; and female participants, n = 3796. Dotted line: reference; solid line: 95% confidential interval; black circle: ABSI Z-score, unadjusted model; black square: ABSI Z-score, fully adjusted model; black triangle pointing upward: BMI Z-score, unadjusted model; black triangle pointing downward: BMI Z-score, fully adjusted model; white circle: ABSI Z-score, unadjusted model; white square: ABSI Z-score, fully adjusted model; white triangle pointing upward: BMI Z-score, unadjusted model; white triangle pointing downward: BMI Z-score, fully adjusted model; half white and half black circle: ABSI Z-score, unadjusted model; half white and half black square: ABSI Z-score, fully adjusted model; half white and half black triangle pointing upward: BMI Z-score, unadjusted model; half white and half black triangle pointing downward: BMI Z-score, fully adjusted model. * *p* < 0.05. ** *p* < 0.01. *** *p* < 0.001 for odds ratio of moderate-to-severe chronic kidney disease, compared with the lowest quartile. Abbreviations: ABSI, A Body Shape Index; BMI, body mass index; MQ, middle quartile; HQ, highest quartile.

Figure 3 shows overall and sex-specific ROC curves pertaining to ABSI for the *S*_CKD_ group. The optimal association between sensitivity and specificity was achieved at ABSI ≥0.0812, ≥0.0822, and 0.0795 in overall, male, and female participants, respectively (*p* < 0.001). Regarding the cutoff points for CKD, sensitivity of 62.92% and specificity of 55.82% (Figure 3(A1)), sensitivity of 56.25% and specificity of 62.68% (Figure 3(A2)), and sensitivity of 74.26% and specificity of 45.17% (Figure 3(A3)), respectively, were observed. The optimal association between sensitivity and specificity was achieved at BMI ≥ 25.86, ≥ 24.96, and 25.87 in overall, male, and female participants, respectively (*p* < 0.01). Regarding the cutoff points for CKD, sensitivity of 35.71% and specificity of 73.14% (Figure 3(B1)), sensitivity of 42.81% and specificity of 66.19% (Figure 3(B2)), and sensitivity of 40.24% and specificity of 69.87% (Figure 3(B3)), respectively, were observed. The AUC yield for ABSI was higher than that for BMI.

Overall participants, n = 7053; male participants, n = 3257; and female participants, n = 3796. Red line: reference; blue line: area under the curve (AUC) that indicates the accuracy of the A Body Shape Index (A) and BMI (B) to detect moderate-to-severe chronic kidney disease (CKD); cutoff point: the value of ABSI that predicts moderate-to-severe CKD; sensitivity: the ratio of individuals who actually have moderate-to-severe CKD and are predicted to have moderate-to-severe CKD; specificity: the ratio of individuals who do not have moderate-to-severe CKD and are not predicted to have moderate-to-severe CKD. Abbreviations: ABSI, A Body Shape Index; BMI, body mass index; MQ, middle quartile; HQ, highest quartile.

## 4. Discussion

To our knowledge, this is the first attempt to explore the association between ABSI and CKD and the predictive capacity of ABSI for CKD compared with that of BMI in a senior population. Our results suggest that ABSI might be a better index than BMI for screening and detecting high-risk individuals with CKD.

Because body composition impairments such as progressive muscle mass loss and fat mass gain with aging are more closely related to various health conditions such as CKD and type 2 diabetes than with BMI, a new anthropometric index that can reflect body composition more precisely than BMI may be helpful in the field [12,25,26]. Therefore, the ABSI, which quantifies abdominal adiposity relative to the BMI and height, might be more suitable to assess the body composition than other anthropometric indexes in the senior population. The ABSI showed outstanding performance in evaluating body composition compared with the WC and BMI in one study of older adults [27] and in identifying visceral obesity and decreased muscle mass in another study [28]. Additionally, the ABSI is reportedly a stronger predictor of type 2 diabetes than is the BMI [21]. Similarly, because the prevalence of hypertension, myocardial infarction, and cardiovascular mortality are linked to detrimental changes in the body composition with aging and are interrelated with CKD, the ABSI would be a very helpful tool to screen and detect high-risk individuals with these health conditions [6,16,29,30,31].

In the present study, for overall, male, and female participants, BMI could distinguish between the N group and M_CKD_ and S_CKD_ groups, although it could not distinguish between the M_CKD_ and S_CKD_ groups. In contrast, ABSI could discriminate between each group (from the N to S_CKD_ group). Additionally, although both ABSI and BMI increased gradually with deterioration in CKD, the strength of an increasing trend in ABSI (SS = 6.48, *p* < 0.001) was comparable to that of BMI (SS = 6.38, *p* < 0.001) in male participants, and was 27.6% and 44.1% higher with ABSI in overall and female participants, respectively, compared with BMI (Table 2). Similarly, the degree of correlation with eGFR in male participants was similar between ABSI and BMI, although in overall and female participants, ABSI was 30.8% and 50.0% higher, respectively, compared with BMI (Table 3). For decades, the BMI-based evaluation of obesity was universally used to investigate the association between obesity and various chronic diseases, and its validity and reliability have been adequately verified [32,33]. However, recent studies indicated that higher fat mass is closely linked to an increased risk of chronic diseases [13,34,35]. Additionally, abdominal fat accumulation is recognized to be incredibly perilous [36,37,38]. Given these recent reports and the findings from the present study, the limitations of BMI become apparent. ABSI, which comprehensively evaluates anthropometric parameters including WC, body weight, and height, is highly relevant.

On the basis of comparisons of overall and sex-specific odds ratios for the associations between Z-scores of ABSI or BMI and *S*_CKD_, Z-scores of ABSI in overall, male, and female participants were higher than those of BMI in the unadjusted and fully adjusted models (Figure 2). However, in both unadjusted and fully adjusted models, the difference between the Z-score of ABSI and BMI was 1.48 and 1.14 times higher in women and 1.10 and 0.77 times higher in men, respectively (Figure 2). These findings indicate that the capacity of ABSI to predict *S*_CKD_ is higher in women than in men.

The sex difference observed to be related to the capacity of ABSI to predict *S*_CKD_ may reflect the fact that the kidneys of women are more sensitive to abdominal fat accumulation than those of men. Similarly, according to Karlsson et al. (2019) and Sorimachi et al. (2021), the associations between visceral fat and hypertension, hyperlipidemia, type 2 diabetes, and heart attack/angina are stronger in women compared with men [39,40]. Regarding type 2 diabetes, which is strongly associated with CKD, a 1-kg increase in abdominal fat increases the risk of type 2 diabetes 2.50-fold and 7.34-fold in men and women, respectively [40]. However, the higher overall prevalence of health conditions in male participants may be the consequence of higher average deposits of visceral fat in men.

AUC analysis (Figure 3) revealed that the AUC yield for ABSI was higher than that for BMI, and ABSI had a significant capacity for predicting CKD. The resulting cutoff points were ≥0.0812 (sensitivity: 62.92% and specificity: 55.82%) in overall participants, ≥0.0822 (sensitivity: 56.25% and specificity: 62.68%) in male participants, and 0.0795 (sensitivity: 74.26% and specificity: 45.17%) in female participants. Because this is the first attempt to provide ABSI cutoff points for CKD, no direct comparisons with previous studies can be made. However, Duncan et al. (2013) reported that the ABSI cutoff point for metabolic syndrome was ≥0.076 in Portuguese male and female adolescents, and that insulin resistance was ≥0.073 and ≥0.079 in female and male participants, respectively [41]. Behboudi-Gandevani et al. (2016) reported that the ABSI cutoff points for metabolic syndrome and insulin resistance were ≥0.077 and ≥0.078, respectively, in young female participants [42]. On the basis of the findings of the present and previous studies, given age, race, and sex differences, an ABSI over 0.073–0.079 appears to be the threshold for various health conditions.

The present study had several strengths and limitations. Important potential covariates such as demographic parameters and lifestyle factors that may influence the association between ABSI and CKD were controlled. Since the present study is a cross-sectional study, it is difficult to strongly confirm the predictive capacity of ABSI for CKD. A longitudinal cohort study needs to be performed to confirm the findings yielded by the present study. Furthermore, the findings of the present study cannot necessarily be translated to other ethnicities or areas outside of Korea because all the study participants were from a Korean senior population. Further studies involving other ethnicities must be performed to generalize the association between the ABSI and CKD. Finally, because we conducted this study solely on an aging population, we deem that it is necessary to conduct a study involving young and middle-aged populations to investigate the independence of the ABSI in the future.

## 5. Conclusions

Compared with the association between BMI and CKD, a similar or stronger association was observed between ABSI and CKD in male or overall and female participants, respectively. However, ABSI had a better capacity than BMI to discriminate between each group (from the *N* to *S*_CKD_ group). Moreover, the capacity of ABSI to predict *S*_CKD_ was higher than that of BMI in overall participants and was more substantial in women than in men. ABSI cutoff points for CKD were ≥0.0812, ≥0.0822, and0.0795 in overall, male, and female participants, respectively. These findings suggest that ABSI is superior to BMI for screening and detecting high-risk individuals with CKD.

## Figures and Tables

**Figure 1 ijerph-18-12874-f001:**
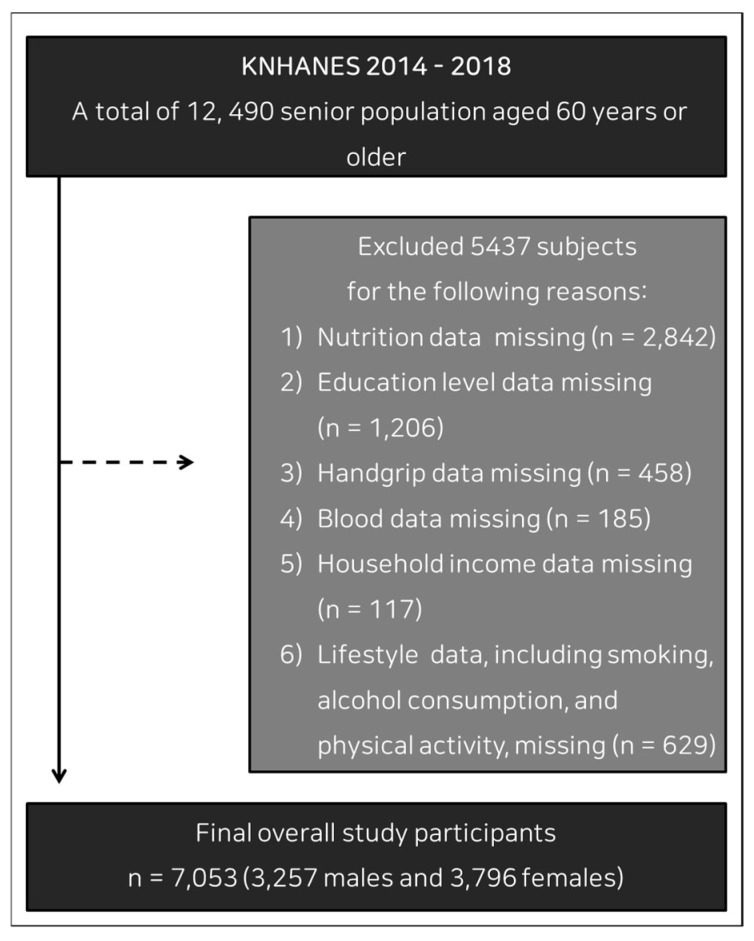
Flow chart of participant recruitment for the study.

**Figure 2 ijerph-18-12874-f002:**
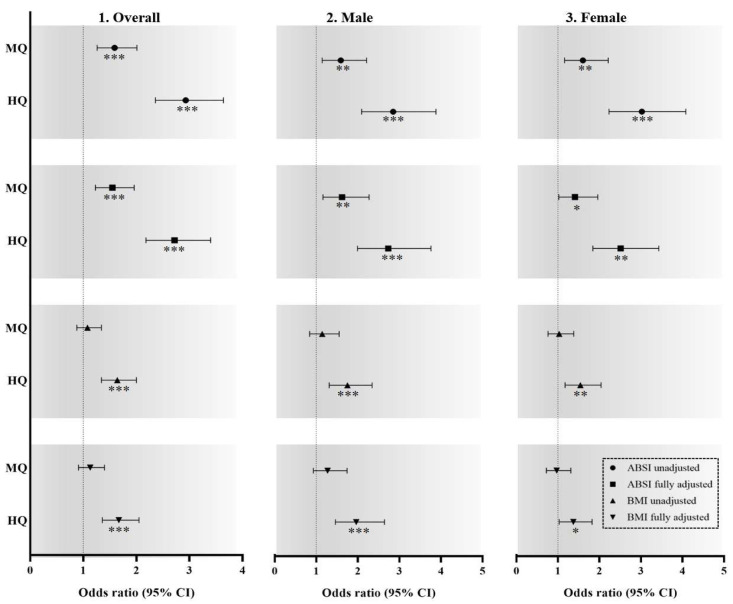
Comparison of overall and sex-specific odds ratios for the association between the Z-score of ABSI or BMI and moderate-to-severe chronic kidney disease. * *p* < 0.05. ** *p* < 0.01. *** *p* < 0.001 for odds ratio of moderate-to-severe chronic kidney disease, compared with the lowest quartile.

**Figure 3 ijerph-18-12874-f003:**
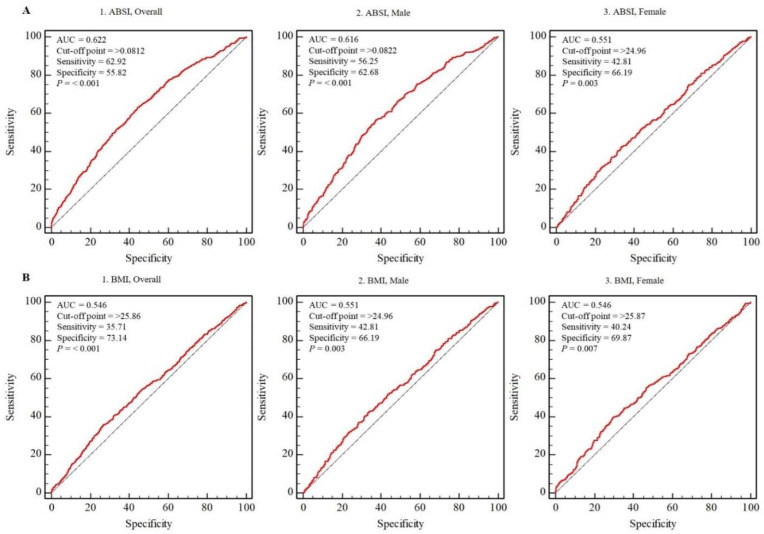
Overall and sex-specific ROC curves of ABSI (**A**) and BMI (**B**) for moderate-to-severe chronic kidney disease.

**Table 1 ijerph-18-12874-t001:** The characteristics of study participants.

	Overall (n = 7053)	Male (n = 3257)	Female (n = 3796)	*p* Value
eGFR, mg/dL	61.0 ± 12.6	60.9 ± 12.5	61.1 ± 12.6	=0.510
Cre, mg/dL ^†^	0.9 ± 0.3	1.0 ± 0.3	0.8 ± 0.3	<0.001
Age, year ^†^	69.5 ± 6.3	69.5 ± 6.2	69.5 ± 6.4	=0.884
Height, cm	159.0 ± 8.7	166.1 ± 5.8	152.9 ± 5.7	<0.001
Weight, kg ^†^	61.4 ± 9.9	66.0 ± 9.5	57.4 ± 8.5	<0.001
WC, cm ^†^	85.0 ± 8.9	86.7 ± 8.6	83.5 ± 8.9	<0.001
BMI, kg/m^2 †^	24.2 ± 3.1	23.9 ± 2.9	24.5 ± 3.3	<0.001
ABSI ^†^	0.0806 ± 0.0042	0.0811 ± 0.0038	0.0802 ± 0.0045	<0.001

Values are means ± SD. ^†^ Mann–Whitney U test was applied to assess the difference between groups. eGFR = estimated glomerular filtration rate; Cre = Creatinine; WC = waist circumference; BMI = body mass index; ABSI = A Body Shape Index.

**Table 2 ijerph-18-12874-t002:** The overall and gender-specific differences, as well as trends of participants, by eGFR category.

	eGFR Category (mL/min/1.73 m^2^)			*p* (Difference)	*SS*	*p* (Trend ^‡^)
	*N* eGFR ≥ 60 (95% CI)	*M*_CKD_ eGFR 45–59.9 (95% CI)	*S*_CKD_ eGFR < 45 (95% CI)			
Overall	n = 3800	n = 2595	n = 658			
eGFR, mg/dL ^†^	69.9 ± 8.1 (69.7, 70.2)	53.8 ± 4.1 (53.7, 54.0)	38.0 ± 7.1 (37.5, 38.6)	A > B > C	−80.01	<0.001
Cre, mg/dL ^†^	0.74 ± 0.13 (0.73, 0.74)	0.92 ± 0.14 (0.92, 0.93)	1.34 ± 0.65 (1.29, 1.39)	A < B < C	57.86	<0.001
Age, year ^†^	68.1 ± 5.9 (67.9, 68.3)	70.5 ± 6.3 (70.3, 70.8)	74.0± 5.7 (73.6, 74.4)	A < B < C	23.22	<0.001
Height, cm ^†^	159.0 ± 8.6 (158.7, 159.3)	159.1 ± 8.7 (158.8, 159.5)	158.4 ± 9.1 (157.7, 159.1)	NS	−0.27	=0.786
Weight, kg	60.7 ± 9.9 (60.4, 61.1)	62.1 ± 9.9 (61.7, 62.5)	62.2 ± 10.3 (61.4, 63.0)	A < B, C	5.89	<0.001
WC, cm	83.9 ± 8.8 (83.6, 84.2)	85.8 ± 8.7 (85.5, 86.1)	87.7 ± 9.2 (87.0, 88.4)	A < B < C	11.41	<0.001
BMI, kg/m^2 †^	24.0 ± 3.1 (23.9, 24.1)	24.5 ± 3.0 (24.4, 24.6)	24.7 ± 3.3 (24.5, 25.0)	A < B, C	7.46	<0.001
BMI Z-Score ^†^	0.02 ± 0.90 (−0.01, 0.05)	0.16 ± 0.88 (0.13, 0.20)	0.24 ± 0.96 (0.16, 0.31)	A < B, C	7.46	<0.001
ABSI ^†^	0.0802 ± 0.0040 (0.0800, 0.0803)	0.0809 ± 0.0043 (0.0807, 0.0810)	0.0822 ± 0.0044 (0.0819, 0.0826)	A < B < C	10.31	<0.001
ABSI Z-Score	0.5717 ± 0.8846 (0.5436, 0.5998)	0.7234 ± 0.9603 (0.6865, 0.7604)	1.0331 ± 0.9766 (0.9584, 1.1079)	A < B < C	10.31	<0.001
Male	n = 1752	n = 1185	n = 320			
eGFR, mg/dL ^†^	69.9 ± 8.0 (69.5, 70.2)	53.8 ± 4.1 (53.6, 54.1)	38.9 ± 12.5 (37.4, 38.9)	A > B > C	−54.58	<0.001
Cre, mg/dL ^†^	0.85 ± 1.08 (0.85, 0.85)	1.06 ± 0.08 (1.06, 1.07)	1.52 ± 0.61 (1.45, 1.59)	A < B < C	23.61	<0.001
Age, year ^†^	68.1 ± 5.8 (67.8, 68.4)	70.6 ± 2.9 (70.2, 70.9)	73.6 ± 5.8 (72.9, 74.2)	A < B < C	15.47	<0.001
Height, cm	166.0 ± 5.9 (165.7, 166.3)	166.2 ± 5.7 (165.9, 166.5)	165.9 ± 5.5 (165.3, 166.5)	NS	0.69	=0.492
Weight, kg	65.2 ± 9.6 (64.7, 65.6)	67.0 ± 9.3 (66.5, 67.6)	67.1 ± 9.3 (66.1, 68.2)	A < B, C	5.76	<0.001
WC, cm	85.5 ± 8.6 (85.1, 85.9)	87.7 ± 8.2 (87.2, 88.1)	89.2 ± 8.6 (88.3, 90.1)	A < B < C	8.56	<0.001
BMI, kg/m^2^	23.6 ± 2.9 (23.5, 23.7)	24.2 ± 2.8 (24.1, 24.4)	24.4 ± 2.9 (24.0, 24.7)	A < B, C	6.38	<0.001
BMI Z-Score	−0.09 ± 0.85 (−0.13, −0.05)	0.09 ± 0.82 (0.05, 0.14)	0.13 ± 0.83 (0.04, 0.22)	A < B, C	6.38	<0.001
ABSI	0.0808 ± 0.0038 (0.0806, 0.0810)	0.0813 ± 0.0038 (0.0811, 0.0815)	0.0825 ± 0.0037 (0.0821, 0.0829)	A < B < C	6.48	<0.001
ABSI Z-Score	0.7053 ± 0.8386 (0.6660, 0.7446)	0.8286 ± 0.8376 (0.7808, 0.8763)	1.0942 ± 0.8261 (1.0033, 1.1850)	A < B < C	6.48	<0.001
Female	n = 2048	n = 1410	n = 338			
eGFR, mg/dL ^†^	70.0 ± 8.2 (69.6, 70.4)	53.8 ± 4.1 (53.6, 54.0)	37.9 ± 7.2 (37.1, 38.7)	A > B > C	−58.50	<0.001
Cre, mg/dL ^†^	0.64 ± 0.06 (0.64, 0.65)	0.81 ± 0.06 (0.80, 0.81)	1.18 ± 0.65 (1.11, 1.25)	A < B < C	57.81	<0.001
Age, year ^†^	68.1 ± 5.8 (67.8, 68.4)	70.5 ± 6.1 (70.2, 70.9)	74.4 ± 5.6 (73.8, 75.0)	A < B < C	17.31	<0.001
Height, cm	153.0 ± 5.6 (152.8, 153.2)	153.2 ± 5.9 (152.9, 153.5)	151.4 ± 5.6 (150.8, 152.0)	A, B > C	−2.45	<0.05
Weight, kg	57.0 ± 8.5 (56.6, 57.3)	58.0 ± 8.3 (57.5, 58.4)	57.5 ± 9.0 (56.6, 58.5)	A < B, C	3.15	<0.01
WC, cm	82.5 ± 8.8 (82.1, 82.9)	84.2 ± 8.7 (83.8, 84.7)	86.2 ± 9.5 (85.2, 87.2)	A < B < C	7.71	<0.001
BMI, kg/m^2 †^	24.3 ± 3.2 (24.2, 24.4)	24.7 ± 3.2 (24.5, 24.8)	25.1 ± 3.7 (24.7, 25.5)	A < B, C	4.43	<0.001
BMI Z-Score ^†^	0.11 ± 0.93 (0.07, 0.15)	0.22 ± 0.93 (0.17, 0.27)	0.34 ± 1.06 (0.23, 0.45)	A < B, C	4.43	<0.001
ABSI ^†^	0.0797 ± 0.0041 (0.0795, 0.0798)	0.0805 ± 0.0047 (0.0802, 0.0807)	0.0820 ± 0.0049 (0.0815, 0.0825)	A < B < C	7.92	<0.001
ABSI Z-Score	0.4574 ± 0.9068 (0.4181, 0.4967)	0.6350 ± 1.0446 (0.5805, 0.6896)	0.9753 ± 1.0985 (0.8578, 1.0928)	A < B < C	7.92	<0.001

Values are means ± SD. ^†^ The Mann–Whitney U test was applied to assess the difference between groups. ^‡^ The Jonckheere–Terpstra test was used to assess the trend among three groups. eGFR = estimated glomerular filtration rate; Cre = creatinine; WC = waist circumference; BMI = body mass index; ABSI = A Body Shape Index; SS = standardized statistic; NS = not significant.

**Table 3 ijerph-18-12874-t003:** The overall and gender-specific Pearson correlation of ABSI and BMI with eGFR.

	Overall		Male		Female	
	ABSI	BMI	ABSI	BMI	ABSI	BMI
eGFR	−0.146 ***	−0.101 ***	−0.130 ***	−0.132 ***	−0.158 ***	−0.079 ***

eGFR = estimated glomerular filtration rate. BMI = body mass index; ABSI = A Body Shape Index. *** *p* < 0.001.

## Data Availability

The datasets analyzed during the current study are available from the corresponding author on reasonable request.

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
