# Peer review of "The A Body Shape Index Might Be a Stronger Predictor of Chronic Kidney Disease Than BMI in a Senior Population"

_ijerph, 2021, doi:10.3390/ijerph182412874_

Round 1

Reviewer 1 Report

Thank you for the opportunity to review this manuscript. In order to improve its technical quality some considerations should be taken into account:

1)Briefly describe why ABSI may be more suitable to asses body mass composition in senior patients.

2) Briefly describe if there were any differences regarding comorbidities (i.e. diabetes and CKD, and if there is any association between ABSI and comorbidities)

3)Briefly discuss how you found out the independence of ABSI from other CKD-related factors, such as age.

Suggested references include;

Krakauer NY, Krakauer JC. Association of Body Shape Index (ABSI) with Hand Grip Strength. Int J Environ Res Public Health. 2020 Sep 17;17(18):6797. doi: 10.3390/ijerph17186797. PMID: 32957738; PMCID: PMC7558329.

Ji M, Zhang S, An R. Effectiveness of A Body Shape Index (ABSI) in predicting chronic diseases and mortality: a systematic review and meta-analysis. Obes Rev. 2018 May;19(5):737-759. doi: 10.1111/obr.12666. Epub 2018 Jan 19. PMID: 29349876.

Bertoli S, Leone A, Krakauer NY, Bedogni G, Vanzulli A, Redaelli VI, De Amicis R, Vignati L, Krakauer JC, Battezzati A. Association of Body Shape Index (ABSI) with cardio-metabolic risk factors: A cross-sectional study of 6081 Caucasian adults. PLoS One. 2017 Sep 25;12(9):e0185013. doi: 10.1371/journal.pone.0185013. PMID: 28945809; PMCID: PMC5612697.

Author Response

We are thankful for the comments and suggestions made by the editor and reviewers, and we have revised the manuscript accordingly

Reviewer 2 Report

It is exciting material in the face of the need to investigate the association of body shape and chronic disease in a senior population. Considering that, I would recommend that:

- Authors should revise the abstract that presents reference indication;

- method needs to present a better explanation of the anthropometric measurements protocols  of the Anthropometric índices;

- conclusion needs to include recommendations of further investigations to overcome the gaps of the presented research.

Author Response

We are thankful for the comments and suggestions made by the editor and reviewers, and we have revised the manuscript accordingly.

Comment 1.

Authors should revise the abstract that presents reference indication

Response to Comment 1:

In accordance with the reviewer’s suggestion, we have revised the abstract and added a reference citation on Page 1, Lines 13–25 as follows: The A Body Shape Index (ABSI) was recently introduced to quantify abdominal adiposity relative to the BMI and height. (Krakauer NY et al., 2012). This cross-sectional study was performed to explore whether the ABSI is linked to chronic kidney disease (CKD) in older adults and compare the predictive capacity of the ABSI versus BMI for CKD. In total, 7,053 people aged ≥60 years were divided into normal, mild, and moderate-to-severe CKD groups based on their estimated glomerular filtration rate (eGFR). The correlation of the ABSI with the eGFR and the differences and trends in the ABSI and BMI among the groups were analyzed, and the cutoff-points for moderate-to-severe CKD were calculated. The association between the ABSI and CKD was stronger than that between the BMI and CKD. The ABSI had a better capacity to discriminate the CKD stage than did the BMI. The capacity of the ABSI to predict moderate-to-severe CKD was higher than that of the BMI and was more substantial in women than men. The ABSI cutoff-points for CKD were ≥0.0822 and 0.0795 in men and women, respectively. In conclusion, the ABSI serves as a better index than the BMI for screening and detecting high-risk individuals with CKD.

Comment 2.

method needs to present a better explanation of the anthropometric measurements protocols of the Anthropometric índices;

Response to Comment 2:

In accordance with the reviewer’s suggestion, we have revised the anthropometric measurements protocols on Page 3, Lines 79–84 as follows: Waist circumstance (WC) and height were assessed to the nearest 0.1 cm. Body weight was assessed to within 0.1 kg with a digital electronic scale (JENIX DS-102; Dong Sahn Jenix Co., Seoul, Korea). Using this information, BMI was defined as body weight (kg) / height (m2), and ABSI was defined as WC × body weight−2/3 × height5/6 [15]. To compare these two anthropometric indexes (BMI and ABSI), the indexes were converted to a Z-score using the following equation: (assessment value − mean) / standard deviation [21].

Comment 3.

conclusion needs to include recommendations of further investigations to overcome the gaps of the presented research.

Response to Comment 3:

In accordance with the reviewer’s suggestion, we have added the following text to the Discussion on Page 9, Lines 302–308: Furthermore, the findings of the present study cannot necessarily be translated to other ethnicities or areas outside of Korea because all the study participants were from a Korean senior population. Further studies involving other ethnicities must be per-formed to generalize the association between the ABSI and CKD. Finally, because we conducted this study solely on an aging population, we deem that it is necessary to conduct a study involving young and middle-aged populations to investigate the in-dependence of the ABSI in the future.

Reviewer 3 Report

I would like to start by congratulating the authors and congratulating them for having decided to investigate an area where there is still so much to discover, but also for having decided to share this findings with the rest of the scientific community, so that science can evolve.

This cross-sectional study is about the discussion about a body shape index that might be a stronger predictor of chronic kidney disease than BMI in a senior population.

All comments, questions and suggestions presented are constructive and try to improve the article, after several careful readings.

Abstract

Objective presented in the abstract must be the same as defined at the end of the Introduction

Keywords

Repetitions with expressions that are in the title should be avoided. Whenever possible, keywords should be Mesh.

Introduction

The objective at the end of the Introduction must be exactly the same as that of the Abstract or vice versa.

The bibliographical review leaves a little to be desired, especially in a topic that has had so much evolution over the years. References from 2003, 2006, 2007, 2012, 2014 (2), 2015, 2016 (2), 2018 (2), 2019 (2), 2020 cannot be considered recent nor justify the current state of the art and are not enough to meet the standards of this magazine. By the way, with these references is the feeling that this article will have already been submitted and rejected and not updated in this component.

Materials and Methods and Results

Figure 1 should have more quality.

Figure 2 and 3 should have more quality.

Discussion and Conclusions

The same note about the bibliographical references, noted above.

General comments

Interesting article, with clear potential for publication, with a very interesting approach and with the potential to change clinical decisions.

The bibliography presented is not in line with the quality of this journal.

The figures presented deserve a careful review.

The article as presented must submitted to major revision.

Author Response

We are thankful for the comments and suggestions made by the editor and reviewers, and we have revised the manuscript accordingly.

Round 2

Reviewer 3 Report

The authors carried out an important review of their article. My opinion is that it should be accepted for publication.
Just a note to Table 3 which should be placed at the top of the next page.